# Evaluation of the Presence of ASFV in Wolf Feces Collected from Areas in Poland with ASFV Persistence

**DOI:** 10.3390/v13102062

**Published:** 2021-10-14

**Authors:** Maciej Szewczyk, Krzysztof Łepek, Sabina Nowak, Małgorzata Witek, Anna Bajcarczyk, Korneliusz Kurek, Przemysław Stachyra, Robert W. Mysłajek, Bogusław Szewczyk

**Affiliations:** 1Department of Vertebrate Ecology and Zoology, Faculty of Biology, University of Gdańsk, Wita Stwosza 59, 80-308 Gdańsk, Poland; maciej.szewczyk@ug.edu.pl (M.S.); witekmalgorzata95@gmail.com (M.W.); 2Department of Recombinant Vaccines, Intercollegiate Faculty of Biotechnology, University of Gdańsk, Abrahama 58, 80-307 Gdańsk, Poland; krzysztof.lepek@ug.edu.pl (K.Ł.); a.bajcarczyk@gmail.com (A.B.); 3Department of Ecology, Institute of Functional Biology and Ecology, Faculty of Biology, Biological and Chemical Research Centre, University of Warsaw, Żwirki i Wigury 101, 02-089 Warszawa, Poland; sabina.pieruzeknowak@gmail.com; 4Masurian Centre for Biodiversity, Research and Education, Faculty of Biology, University of Warsaw, Urwitałt 1, 11-730 Mikołajki, Poland; kornel.kurek@gmail.com; 5Roztocze National Park, Plażowa 2, 22-470 Zwierzyniec, Poland; monitoring@roztoczanskipn.pl

**Keywords:** ASF, gray wolf, wild boar, virus transmission

## Abstract

African swine fever (ASF), caused by a DNA virus (ASFV) belonging to genus *Asfivirus* of the *Asfarviridae* family, is one of the most threatening diseases of suids. During last few years, it has spread among populations of wild boars and pigs in countries of Eastern and Central Europe, causing huge economical losses. While local ASF occurrence is positively correlated with wild boar density, ecology of this species (social structure, movement behavior) constrains long-range disease transmission. Thus, it has been speculated that carnivores known for high daily movement and long-range dispersal ability, such as the wolf (*Canis lupus*), may be indirect ASFV vectors. To test this, we analyzed 62 wolf fecal samples for the presence of ASFV DNA, collected mostly in parts of Poland declared as ASF zones. This dataset included 20 samples confirmed to contain wild boar remains, 13 of which were collected near places where GPS-collared wolves fed on dead wild boars. All analyzed fecal samples were ASFV-negative. On the other hand, eight out of nine wild boar carcasses that were fed on by telemetrically studied wolves were positive. Thus, our results suggest that when wolves consume meat of ASFV-positive wild boars, the virus does not survive the passage through intestinal tract. Additionally, wolves may limit ASFV transmission by removing infectious carrion. We speculate that in areas where telemetric studies on large carnivores are performed, data from GPS collars could be used to enhance efficiency of carcass search, which is one of the main preventive measures to constrain ASF spread.

## 1. Introduction

African swine fever (ASF) is a viral disease of suids that causes high mortality (nearly 100% in domestic pigs) [1]. The disease is caused by African swine fever virus (ASFV), classified as the only member of the *Asfivirus* genus, within the *Asfarviridae* family [1,2]. It is a large (average diameter 200 nm), enveloped virus with icosahedral morphology. ASFV genome is a single molecule of linear double-stranded DNA, ranging from 170 up to 190,000 base pairs in length, encoding over 150 open reading frames [1,2].

ASF naturally occurs in sub-Saharan Africa, where it is transmitted by soft ticks of the *Ornithodoros* genus. It is the only DNA virus transmitted by arthropods [1]. Among its natural hosts, there are also African suids such as warthogs, giant forest hogs, and bush pigs that develop asymptomatic infections. These animals are considered as a natural reservoir of the virus in Africa [3].

In Europe, ASF was introduced to the Iberian Peninsula in 1957, and later spread into some other European countries, but was nearly eradicated from Europe by 1990 due to drastic control and eradication programs, remaining only in Sardinia, where it became endemic. However, in 2007, it was introduced to the Caucasus and subsequently expanded through the Russian Federation, Ukraine, and Belarus until it eventually entered the European Union in 2014, where it has been spreading throughout Poland and the Baltic countries [4]. Recently it was detected also west of the Polish border, e.g., in eastern Germany [5].

The wild boar (*Sus scrofa*) has been implicated in the ASF epidemiology in Europe. It has been shown that the disease can persist in wild boar populations without spillover from domestic pigs or other sources of infection [4,6]. However, the role of wild boar in ASFV spread is not fully understood. Wild boar density and proportion of forested area were found to be positively correlated with the probability of detecting ASF cases [7], but several factors limit direct disease transmission, most importantly wild boar ecology (small home ranges, social structure, relatively short dispersal range) and the severity of the disease, which quickly hampers movement ability of infected individuals [8,9]. Thus, the main way of local ASF spread among wild boars may be indirect transmission through infected carcasses [8,10,11]. ASFV is known for its high resistance to environmental conditions [12], e.g., the virus remained infectious in infected spleen samples buried in soil for 280 days [13].

As direct long-range disease spread by wild boars is unlikely due to reasons described above, new isolated ASF outbreaks in free-ranging wild boars are often attributed to anthropogenic, indirect disease transmission, e.g., via contaminated meat, behavior of hunters (e.g., poor biosecurity measures during transport of killed game), or through fomites (contaminated clothing, equipment) [4,14,15,16]. However, it has been also speculated that ASFV could be transmitted once wild boar carcasses are eaten by other carnivores or birds [17]. While a recent study indicated that scavengers represent a minor risk factor for spreading ASF [18], numerous articles in Polish popular and agricultural press suggest a role of wolves in the long-range disease spread (e.g., [19]). Such undocumented speculations could affect public attitude towards wolves and have negative effect on wolf conservation in Poland.

Wolves were nearly exterminated from Central Europe but have recently recolonized a large part of the species’ historical range [20]. In Poland, since strict species protection was implemented in 1998, the wolf population has been dynamically expanding and wolves recolonized most of the suitable habitats, including vast forest tracts west of the Vistula River [21,22,23,24]. Wild boar is an important food item for wolves—in western Poland, where wild boar densities are the highest [25], this species constituted 23 to 43% of biomass consumed by wolves [26]. It was found that ASF-driven decrease of wild boar numbers affects wolf feeding ecology—after the wild boar population in Vitebsk region (NE Belarus) declined, wolves started to hunt roe deer and red deer that were earlier avoided by local wolf packs [27].

While active transmission of ASFV by wolves could be ruled out, as the only virus’ mammal hosts are suids [14], according to our best knowledge, the possibility of indirect virus spread through feces of wolves that fed on infected boars has not been verified. Performing an experimental test investigating if ASFV could remain infectious after passaging through the wolf intestinal tract would be challenging due to biosecurity and ethical concerns. Therefore, we tried to explore this problem indirectly by detecting ASFV genetic material in non-invasively collected wolf scats. Taking an advantage of GPS-GSM telemetry, we localized carcasses of wild boars that likely died of ASF and were subsequently consumed by wolves, and then we analyzed wolf fecal samples collected in the vicinity of the carcasses.

## 2. Materials and Methods

### 2.1. Wolf Telemetry and Sample Collection

Two adult male wolves—M1 in Roztocze National Park (SE Poland) and M2 in the Masurian Lakeland (NE Poland)—were equipped with GPS/GSM collars (Vectronic Aerospace, Berlin, Germany). We live-trapped M1 with a leg-hold trap EZGrip#7 (Livestock Protection Co., Alpine, TX, USA) in January 2020 within a frame of the research project. The individual was a member of a family group consisting of three adult individuals. M2 was found in December 2020 trapped in snares set up by a poacher, and as its hind leg was injured and required surgery; thus, it was kept in the wildlife rehabilitation center until February 2021, when it was released back into its natal home-range. This individual, however, did not join its natal family group but roamed apart. Permission for wolf capturing and marking was granted by the General Directorate for Environmental Protection, while research procedures were approved by the first Local Ethical Commission in Warsaw, Poland.

We set up the GPS schedules in collars to record the locations of studied wolves every one or two hours, but changes in satellite reception caused by weather or terrain conditions meant that not every attempt to localize wolves was successful. We calculated bimonthly (February–March 2021) home-range of both individuals using minimum convex polygon (MCP) with 100%, 95%, and 50% locations (M1 *n* = 608 and M2 *n* = 1201 locations). Calculations were made in R package adehabitatHR [28]. Territories of both wolves were situated in areas declared as ASF zones (Figure 1 and Appendix A). We searched for remains of prey and scats of both GPS-collared individuals throughout the winter 2021 visiting clusters of at least two locations obtained consequently within 100 m and following wolf tracks on snow. We collected wolf scats and tissue samples from prey remains. When prey remains were found, we examined if they were consumed by wolves by snow tracking and searching for wolf bite marks.

Additionally, we analyzed wolf fecal samples collected from 2018 to 2020 during monitoring activities and research projects. We followed wolf tracking and sample preservation methodology described earlier [23,29]. Detailed information on analyzed samples can be found in Appendix A.

### 2.2. Laboratory Analyses

To preselect wolf fecal samples containing wild boar remains, we followed methods described earlier [26]. Briefly, prey species in scats were identified by bone fragments, hooves, claws, feathers, and hair, according to a taxonomic key [30] and by comparison to the reference material of Department of Ecology, University of Warsaw, and Department of Vertebrate Ecology and Zoology, University of Gdańsk.

Both scat and tissue samples were stored in 96% ethanol at 4 °C. DNA from fecal samples was isolated using DNeasy PowerFaecal Pro Kit (Qiagen, Hilden, Germany), while isolation from tissues was performed with Exgene™ Tissue SV kit (GeneAll Biotechnology, Seoul, Korea) according to the manufacturer’s protocols.

To detect ASFV genetic material, we amplified a 257 bp fragment of viral *VP72* gene using primers PPA-1 and PPA-2 [31]. Amplification was performed in 25 µL reactions containing 1 × PCR Mix Plus mixture (A&A Biotechnology, Gdynia, Poland), with each primer at 0.2 µM concentration, 0.2 mg/mL BSA, and 3 µL of DNA isolate. PCR conditions were as follows: initial denaturation for 3 min in 95℃, followed by 35 cycles of 94 °C (15 s), 62 °C (30 s), and 72 °C (45 s), and a final elongation for 7 min at 72 °C.

As a positive control, we used a 315 bp synthetic DNA containing 257 bp fragment identical to the amplified region of ASFV VP72 gene. Synthetic DNA was cloned to pJet1.2/blunt vector (ThermoScientific, Waltham, MA, USA) and propagated in *E. coli* TOP10 cells, followed by plasmid isolation using Plasmid Mini kit (A&A Biotechnology). For details, see the Appendix A.

PCR products were resolved by electrophoresis in 2% agarose gels in 0.25 × TBE buffer and visualized using MidoriGreen reagent. If bands at expected size (257 bp) were visible, PCR products were purified using EppicFast kit (A&A Biotechnology) and sequenced on an ABI3730/xl sequencer (Applied Biosystems) in Genomed S.A. laboratories (Warsaw, Poland). Obtained sequences were examined by eye in FinchTV [32] and compared to NCBI database using BLAST.

## 3. Results

First, we analyzed 32 scat samples, randomly selected from a dataset of samples collected in 2018–2019 in ASF zones. As an additional control, we added to the analysis 10 samples collected in 2018 in western Poland, where ASF was not present at that time (Figure 1 and Appendix A, Appendix A). All these samples were analyzed earlier in 13 canine microsatellite loci and were confirmed to contain wolf DNA. All samples were ASFV-negative, as we observed no amplification products in any sample, while positive controls provided clear bands of expected size (Appendix A). Additionally, we performed a control experiment where a fragment of ASFV genetic material was added to a fresh wolf scat sample, proving that it can be successfully isolated from fecal material (Appendix A, see details in the Appendix A).

Next, we analyzed 20 wolf scats that were confirmed by dietary analysis to contain wild boar remains, all collected in ASF zones. That dataset included 13 very fresh scats found in telemetric locations of monitored wolves (M1 and M2—six and seven samples, respectively; see Figure 1B,C), as well as 7 samples collected during wolf monitoring activities in other parts of eastern Poland (Figure 1A, Appendix A). We did not detect ASFV genetic material in any sample (Figure 2).

In parallel, we analyzed nine DNA samples from remains of wild boar carcasses found in clusters of telemetric locations of studied wolves (Figure 1B,C, Appendix A) and were consumed by them, as indicated by snow tracking. All but one samples turned out to be ASFV-positive, as indicated by agarose gel electrophoresis (Figure 2). This result was confirmed by sequencing of amplification products. All samples had the same sequence and were assigned to the Georgian ASFV genotype that was also reported previously, e.g., in Poland, Germany, Belgium, Czechia, and Estonia (Appendix A).

## 4. Discussion

It has been suggested both in scientific [17] as well as in popular literature (e.g., [19]) that carnivorous mammals such as wolves may be indirect ASFV vectors. Our results do not support this hypothesis. We did not detect ASF DNA in any of 52 wolf fecal samples collected in ASF zones. Moreover, 13 of those samples were very fresh scats that contained wild boar remains and were collected in the near vicinity of locations where wolves were feeding on ASFV-positive wild boar carcasses, yet viral genetic material was not detectable in those scats. Hence, we suggest that passage through wolf intestinal tract leads to decomposition of viral particles and shearing of viral DNA. It was already suggested that upon ingestion, ASFV is very unlikely to remain infectious after passaging through the intestinal tract of a vertebrate [18,33], but according to our best knowledge, this possibility has never been tested experimentally. Ideally, such an experiment could be performed by feeding wolves with ASFV-infected meat in strictly controlled areas, e.g., in zoological gardens and analyzing samples collected at different time points. While due to biosecurity and ethical concerns we did not perform such tests in a controlled environment, our field study mimics such an experiment, as telemetric data and snow tracking clearly indicated that both monitored wolves consumed wild boar carcasses that were confirmed as ASFV-positive. Moreover, in the case of collared individuals, we can estimate the time of feeding, and from the analysis of feces, we can obtain the genetic profile of a particular individual, confirming that the collected scat belonged to the collared individual. Thus, to some extent we can estimate the time of defecation, provided that the collection of fecal samples is done on a regular basis by our ecological staff. As we were not able to detect ASFV DNA in any fecal sample, including those collected probably less than 24 h after defecation, the possibility that infectious ASFV particles may survive in carnivore scats is even less probable.

Our results indicate that wolves are very unlikely to be vectors for long-range ASF spread. However, we cannot exclude that carnivores, including wolves, may sometimes spread the virus in other ways, e.g., via contaminated fur or by dispersing parts of infectious carcasses. Foxes and raccoon dogs were observed ripping parts wild boar carcasses into pieces and carrying them away [18]. Caching behavior was also reported in American wolves [34]. However, this behavior is restricted to short-range transport of food within the animal’s home range, e.g., by parents providing food to their offspring, so it could not be responsible for long range disease spread. Moreover, mammals usually groom their fur when it gets dirty with blood or other body fluids, thus reducing the possibility of virus transfer via contaminated fur.

On the other hand, scavenging carnivores may reduce the risk of virus persistence in the environment, as they efficiently remove wild boar carcasses. It is especially important in the cold season—while in summer, most of the carcass biomass is rapidly decomposed by invertebrates, in winter, mammal and avian scavengers play a major role in the carcass decomposition process [18]. Wolves consume on average 2.8 kg of meat per day [35]. Thus, they may be more efficient in carcass removal than mesocarnivores such as red foxes and racoon dogs, previously reported as the mammal species most frequently scavenging on wild boar carrion in an area of Germany where wolves are not present [18]. In our study, wild boar carcasses found in telemetric locations of studied wolves were reduced to bones and pieces of skin. As indirect virus transmission through infected carrion is thought to be one of the main ways of ASF spread in wild boar populations [8,10,11], we suggest that wolf scavenging on wild boar carcasses is an important factor reducing the risk of ASFV transmission.

Control, eradication, and prevention are the main ways of constraining ASF spread, as no vaccine is available against ASFV [36]. Finding and removing carcasses is one of such crucial measures for effective ASF control, but it is a demanding task, especially in vast forest complexes or marshlands. Thus, several methods to increase efficiency of carcass search teams could be used, such as the use of trained dogs and drones [5] or statistical models predicting selection of deathbed sites by ASFV-infected wild boars [11]. Our results show that data from telemetric studies of large carnivores may be helpful in carcass search, especially in case of carnivore species that have large home range sizes and long daily movement distances, such as brown bears and wolves [28,37,38]. Such application of telemetry may not be a routine method due to relatively small number of large carnivore telemetry projects in Central Europe. However, when ASF outbreaks in wild boar occur in home ranges of telemetrically studied large carnivores, data from GPS collars could be used to assist carcass search teams in their task. Thus, communication between veterinary inspectors responsible for ASF prevention and scientists performing telemetry projects could locally enhance efficiency of constraining ASF spread.

## Figures and Tables

**Figure 1 viruses-13-02062-f001:**
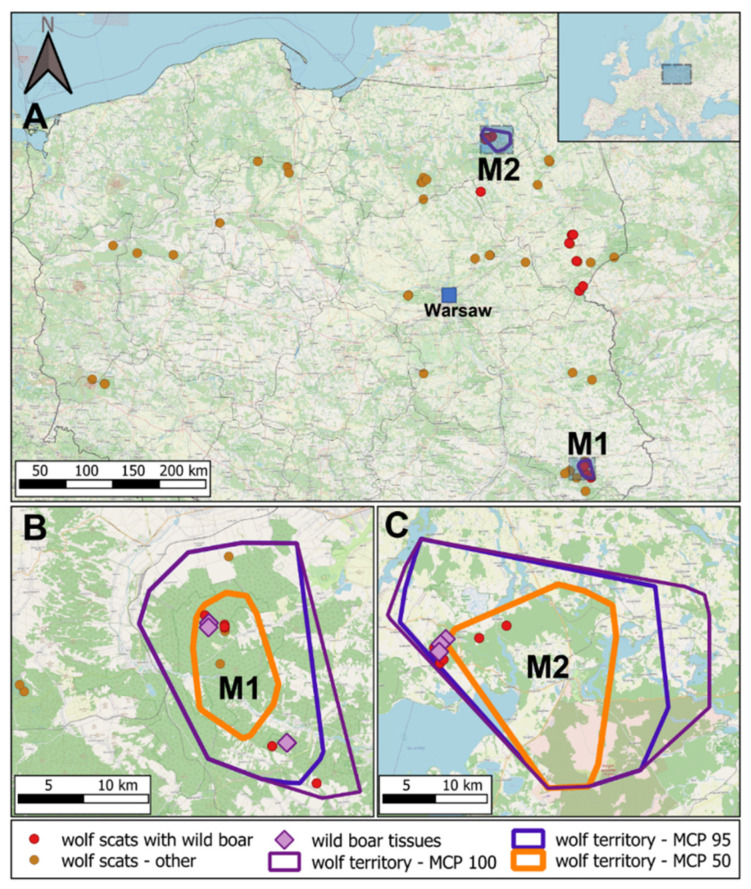
Map of the study area. Upper panel (**A**)—map of Poland (except of southernmost part), showing localization of analyzed wolf fecal samples. Localization of the study area on the map of Europe is indicated in the up right corner. Lower panels—close-up of the areas where telemetric studies of wolves M1 ((**B**) south-east Poland) and M2 ((**C**) north-east Poland) were performed. Calculated minimum convex polygons containing 100% (MCP 100, violet), 95% (MCP 95, blue), or 50% (MCP 50, orange) of telemetric locations are shown. Localizations of wild boar carcasses are indicated with violet diamonds, wolf scats containing wild boar remains are indicated with red dots, and wolf scats for which prey species identity was not analyzed are indicated with orange dots.

**Figure 2 viruses-13-02062-f002:**
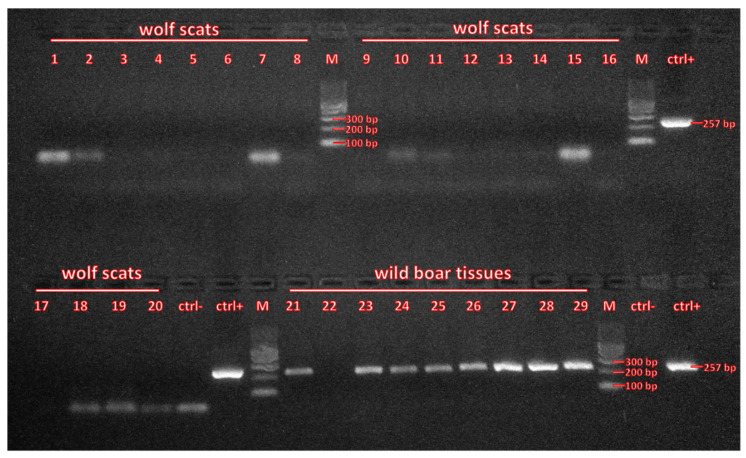
ASFV genetic material was not detectable in wolf fecal samples, even if the carnivores consumed ASFV-positive wild boar carcasses. Image shows PCR amplification products resolved in 2% agarose gel. Lanes numbered 1–20—samples from wolf scats containing wild boar remains (2–8—collected within wolf M2 home range, 9–14—M1 home range, 1 and 15–20—in other areas of eastern Poland where ASF is present), lanes 21–29—samples from wild boar carcasses found in clusters of telemetric locations of studied wolves (21–25—M1, 26–29—M2). Ctrl- represents PCR-negative control (elution buffer from used DNA isolation kits), ctrl+ represents PCR-positive control (plasmid DNA containing the amplified fragment of ASFV VP72 gene). Detailed descriptions of wolf scat and wild boar samples are provided in Appendix A and Appendix A, respectively.

## Data Availability

Detailed information on the analyzed samples can be found in the online Appendix A. Any additional data are available on request from the corresponding author.

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
