# Peer review of "Evaluation of the Presence of ASFV in Wolf Feces Collected from Areas in Poland with ASFV Persistence"

_viruses, 2021, doi:10.3390/v13102062_

Round 1
Reviewer 1 Report
I reviewed the manuscript entitled “Wolves – passive vectors of ASFV or opportunistic carnivores reducing the risk of virus spread? “ . In this communication authors evaluated the potential role of wolfs to excrete ASFV in feces, in regions of Poland where ASFV activity has been detected.
The results presented by the authors are consistent with previous information from the Friedrich-Löffler Institute (Federal Research Institute for Animal Health) in Germany (https://www.openagrar.de/servlets/MCRFileNodeServlet/openagrar_derivate_00009612/FLI-Information_FAQ_ASP20170115.pdf), suggesting that wolfs and other predators are not playing a role in the dissemination of ASF.
Based on the nature of the study presented herein, I suggest authors to change the title for something more specific like: Evaluation of the presence of ASFV in feces of wolfs collected from areas in Poland with ASFV activity.
One criticism to the methodology of this study is the use of conventional PCR instead of real time PCR, a method that might have increased the diagnostic sensitivity (please include in the discussion). Was viral isolation performed on positive PCR samples? Considering the nature of this study, I consider important to include this information. A positive viral isolation may serve as an additional ‘positive control” for this study.
How dietary analysis was conducted?
Another recommendation for the authors to improve quality of this manuscript is to include in the discussion more information regarding the limitations of this study. Since one of the main limitations is of this study is the inability to control the time of sampling collection after feeding with infected carcasses please speculate about ideal experimental studies that would have to be conducted to support the role of wolfs as passive vectors of ASFV.
Reviewer 2 Report
The manuscript by Szewczyk et al. addresses one of the research gaps for ASF transmission, i.e. the role of carnivores as passive vectors of ASFV. The manuscript is well written and very comprehensive. Therefore, I recommend the manuscript to be accepted after minor revision.
Comments
Introduction
Lines 38-39: The sentence is incomplete. It should read “The disease is caused by African Swine Fever Virus (ASFV), classified as the only member of the Asfivirus genus, within the Asfarviridae family [1, 2].” Please also consider the correct spelling of the family name Asfarviridae throughout the manuscript.
Line 60: It should read “cases” instead of “case”.
Line 88: Please adhere to the correct order of the references.
Results
Line 182: The double word “in” should be removed.
Round 2
Reviewer 1 Report
Thank you, the authors for your responses to my comments. I consider that the manuscript was improved and I don’t have any objection to recommend its publication.